# Fairness Constrained Sampling using Augmented Lagrangian Langevin Monte Carlo

## Abstract

Machine learning models trained on biased data can produce systematically un-
fair decisions; this problem is especially acute when we wish to sample from a
Bayesian posterior while enforcing population-level statistical constraints. In this
work we introduce Augmented Lagrangian Langevin Monte Carlo (AL-LMC), a
practical, primal–dual sampler that enforces expectation-type constraints directly
on the posterior distribution rather than on individual samples. Our approach
builds an augmented Lagrangian over probability measures using a Fenchel-Young
reformulation and derives a separable Gibbs tilt whose potential remains tractable
for Langevin updates. Empirically, AL-LMC enforces fairness in Bayesian models,
reducing disparities with little accuracy loss.

## 1 Introduction

Sampling is crucial in statistics and increasingly in machine learning (ML) for uncertainty quan-
tification and generative tasks Faulkner and Livingstone [2024], van de Schoot et al. [2021], Song
and Ermon [2019]. Often, target distributions (e.g., score functions, Bayesian posteriors) are known
only up to a normalization constant. Markov Chain Monte Carlo (MCMC) algorithms, particularly
Langevin Monte Carlo (LMC), address this, gaining attention for their simplicity and effectiveness
Roberts and Rosenthal [2004], Roberts and Tweedie [1996], Song and Ermon [2019], Wibisono
[2018], Durmus et al. [2019], Wang and Li [2022]. However, these algorithms lack natural mech-
anisms for incorporating sample requirements like conditional probabilities for fairness. Rate
constraints have garnered attention in ML due to their central role in fairness Kearns et al. [2018],
Cotter et al. [2019]. Current solutions which include post-processing, variable transformations, or
target distribution penalties, have significant drawbacks. Post-processing (e.g., rejection sampling
Lang et al. [2007], Li and Ghosh [2015]) can severely reduce effective sample size and variable
transformations (e.g., link functions, projections, mirror/proximal maps Hsieh et al. [2018], Bubeck
et al. [2018], Salim and Richtarik [2020], Ahn and Chewi [2021], Kook et al. [2022], Sharrock et al.
[2023], Noble et al. [2023]) are limited to deterministic support constraints and are unsuitable for
statistical requirements like robustness or fairness Madry et al. [2019], Kearns et al. [2018], Cotter
et al. [2019], Chamon et al. [2023]. Direct modification of the target distribution, while flexible
Gürbüzbalaban et al. [2022], does not guarantee constraint satisfaction.

Our main contributions are:

- Propose *Augmented Lagrangian Langevin Monte Carlo* (AL-LMC), a sampler that runs
  Langevin updates on the surrogate while performing stochastic dual updates.

- Empirically show the effectiveness in enforcing statistical parity with minor accuracy loss.

Submitted to 39th Conference on Neural Information Processing Systems (NeurIPS 2025). Do not distribute.

## 2   Fairness Constrained Sampling

Consider data pairs $(\mathbf{x}, y)$, where $\mathbf{x} \in \mathcal{X}$ are features and $y \in \{0, 1\}$ labels, and for $i \in I$ protected (measurable) subgroups $\mathcal{G}_i \subset \mathcal{X}$. Let $\pi$ be a Bayesian posterior of the parameters $\theta$ of a model $q(\cdot; \theta)$ denoting the probability of a positive outcome (based, e.g., on a binomial model) and $\rho_\mathcal{X}$ denote the distribution of features $x$. We use a statistical-parity constraint ensuring group positive rates are not much lower than the population rate. For statistical parity to hold, the probability of a positive outcome must be independent of the protected attribute as in Chamon et al. [2023]. We recast the fairness-constrained sampling problem as

$$
\begin{aligned}
P^\star = \min_{\mu \in \mathcal{P}_2(\mathbb{R}^d)} \quad & \mathrm{KL}(\mu \| \pi) \\
\text{subject to} \quad & \mathbb{E}_{x \sim \rho_\mathcal{X}, \, \theta \sim \mu}[\, q(x; \theta) \mid \mathcal{G}_i \,] \geq \mathbb{E}_{x \sim \rho_\mathcal{X}, \, \theta \sim \mu}[\, q(x; \theta) \,] - \delta_i,
\end{aligned}
\tag{1}
$$

where $\mathcal{P}_2(\mathbb{R}^d)$ is the set of probability measures on $\mathbb{R}^d$ with bounded second moments. Note that this is a statistical constraint and is distinct from prior support-constrained sampling problems considered in Hsieh et al. [2018], Bubeck et al. [2018], Salim and Richtarik [2020], Ahn and Chewi [2021], Kook et al. [2022], Sharrock et al. [2023], Noble et al. [2023], as it constrains the distribution $\mu$ itself, not just its samples $x$. Defining the group probability $p_i = \rho_\mathcal{X}(\mathcal{G}_i) = \Pr_{x \sim \rho_\mathcal{X}}(x \in \mathcal{G}_i)$, we can express the conditional constraint in (1) as linear expectation constraint over $(x, \theta)$ as

$$
\tilde{g}_i(x, \theta) \;=\; p_i \, q(x; \theta) \;-\; q(x; \theta) \, \mathbf{1}_{x \in \mathcal{G}_i} \;-\; p_i \, \delta_i.
$$

The constraint in (1) is therefore equivalent to the simpler form

$$
\mathbb{E}_{\theta \sim \mu}\big[ \bar{g}_i(\theta) \big] \leq 0, \qquad i \in I,
$$

where $\bar{g}_i(\theta) = \mathbb{E}_{x \sim \rho_\mathcal{X}}\big[ \tilde{g}_i(x, \theta) \big]$ is the expected constraint for parameter $\theta$. The primal (distributional) constraint can be written as $\mathbb{E}_{\theta \sim \mu}[\bar{g}(\theta)] \leq 0$ with $\bar{g}(\theta) = (\bar{g}_i(\theta))_{i \in I}$. Algorithms like projections or mirror maps are ill-suited for these types of constraints. We propose a Lagrangian MC algorithm, a sampling analogue of gradient descent-ascent (GDA). Unlike Liu et al. [2021] that required exact expectation computations, this approach overcomes this limitation and provides convergence guarantees for a broader class of constraint functions Chamon et al. [2024]. Hence, constrained sampling provides a natural way to encode fairness in Bayesian inference.

## 3   Augmented Lagrangian Monte Carlo

Throughout this section we use $\theta \in \mathbb{R}^d$ for model parameters and keep the vector of constraint functions $g : \mathbb{R}^d \to \mathbb{R}^I$ from (1). Let $\tilde{\pi}$ denote the (unnormalized) target posterior density and define its potential $f(\theta) := -\log \tilde{\pi}(\theta) + \text{const}$.

**Assumption.** The following assumptions hold throughout this section:

P1. The potential function $f$ has $L_f$-Lipschitz gradient and is $m$-strongly convex outside a ball of radius $R$.

P2. Each constraint function $g_i \in C^1$ with $\nabla g_i$ Lipschitz continuous and $\|\nabla g_i(\theta)\| \leq L_g(1 + \|\theta\|)$ for all $\theta \in \mathbb{R}^d$.

P3. The Slater condition holds: there exists $\theta_0 \in \mathbb{R}^d$ such that $g_i(\theta_0) < 0$ for all $i \in I$.

We define the normalized probability density as $\pi(\theta) = \tilde{\pi}(\theta)/Z$ where $Z = \int \tilde{\pi}(\theta) d\theta$ is the normalization constant. Our goal is not to sample from $\pi$ itself, but from a distribution $\mu^\star$ close to $\pi$ that satisfies the rate constraints in (1). We begin from the augmented Lagrangian over probability measures

$$
\mathcal{L}_\rho(\mu, \lambda) \;=\; \mathrm{KL}(\mu \| \pi) \;+\; \lambda^\top \mathbb{E}_\mu[g] \;+\; \frac{\rho}{2} \big\| \mathbb{E}_\mu[g] \big\|_2^2, \qquad \lambda \in \mathbb{R}_+^I, \; \rho > 0,
\tag{2}
$$

and its dual function

$$
d_\rho(\lambda) \;:=\; \inf_{\mu \in \mathcal{P}_2(\mathbb{R}^d)} \mathcal{L}_\rho(\mu, \lambda).
\tag{3}
$$

We use the Fenchel-Young representation, valid for any $u \in \mathbb{R}^I$,

$$\frac{\rho}{2}\|u\|_2^2 \;=\; \sup_{v \in \mathbb{R}^I} \left\{ v^\top u - \frac{1}{2\rho}\|v\|_2^2 \right\}. \tag{4}$$

Applying (4) with $u = \mathbb{E}_\mu[g]$ in (2) yields

$$d_\rho(\lambda) = \inf_\mu \sup_v \left\{ \mathrm{KL}(\mu\|\tilde{\pi}) + (\lambda + v)^\top \mathbb{E}_\mu[g] - \tfrac{1}{2\rho}\|v\|_2^2 \right\}. \tag{5}$$

By convexity in $\mu$, concavity in $v$, we can swap $\inf$ and $\sup$ using the minimax theorem, obtaining

$$d_\rho(\lambda) \;=\; \sup_{v \in \mathbb{R}^I} \left\{ -\frac{1}{2\rho}\|v\|_2^2 \;+\; \inf_\mu \left( \mathrm{KL}(\mu\|\tilde{\pi}) + (\lambda + v)^\top \mathbb{E}_\mu[g] \right) \right\}. \tag{6}$$

Applying Donsker-Varadhan gives us

$$\inf_\mu \left\{ \mathrm{KL}(\mu\|\tilde{\pi}) + \mathbb{E}_\mu[\phi] \right\} \;=\; -\log \mathbb{E}_\pi \big[ e^{-\phi(\theta)} \big], \quad \text{attained at } \mu_\phi(\mathrm{d}\theta) = \frac{\pi(\theta)\, e^{-\phi(\theta)}}{Z(\phi)} \, \mathrm{d}\theta. \tag{7}$$

Applying (7) to $\phi(\theta) = (\lambda + v)^\top g(\theta)$ and substituting into (6) gives the *concave* dual

$$d_\rho(\lambda) = \sup_{v \in \mathbb{R}^I} \left\{ -\frac{1}{2\rho}\|v\|_2^2 \;-\; \log \mathbb{E}_\pi \big[ e^{-(\lambda+v)^\top g(\theta)} \big] \right\}. \tag{8}$$

Let

$$F(\xi) \;:=\; -\log \mathbb{E}_\pi \big[ e^{-\xi^\top g(\theta)} \big], \qquad \xi \in \mathbb{R}^I, \tag{9}$$

so that $d_\rho(\lambda) = \sup_v \{ -\frac{1}{2\rho}\|v\|_2^2 + F(\lambda + v) \}$. We discuss the properties of $F(\xi)$ and $d_\rho(\lambda)$ in Appendix A and B respectively. The inner $\mu$-minimizer in (7) is the Gibbs tilt

$$\mu_{\lambda+v}(\mathrm{d}\theta) \;\propto\; \pi(\theta) \exp\big( -(\lambda+v)^\top g(\theta) \big) \, \mathrm{d}\theta, \quad \text{i.e.,} \quad U(\theta; \lambda, v) = f(\theta) + (\lambda + v)^\top g(\theta). \tag{10}$$

Thus the sampling potential is *pointwise separable in $\theta$* and contains no quadratic penalty. Writing $\xi = \lambda + v$, we define the function

$$J(\lambda, v) = -\frac{1}{2\rho}\|v\|_2^2 + F(\lambda + v), \tag{11}$$

which has gradients

$$\nabla_v J(\lambda, v) = -\frac{1}{\rho} v + \mathbb{E}_{\mu_{\lambda+v}}[g(\theta)], \qquad \nabla_\lambda J(\lambda, v) = \mathbb{E}_{\mu_{\lambda+v}}[g(\theta)], \tag{12}$$

where $\nabla F(\xi) = \mathbb{E}_{\mu_\xi}[g(\theta)]$ from the properties of the log partition function. By Danskin's theorem, the gradient of the dual function is

$$\nabla_\lambda d_\rho(\lambda) = \nabla_\lambda J\big(\lambda, v^\star(\lambda)\big) = \mathbb{E}_{\mu_{\lambda+v^\star(\lambda)}}[g(\theta)], \tag{13}$$

where $v^\star(\lambda)$ is the unique maximizer of $J(\lambda, \cdot)$ (uniqueness follows from the $-\frac{1}{2\rho}\|v\|_2^2$ term). At any maximizer $(\lambda^\star, v^\star)$ with $\lambda^\star \geq 0$,

$$v^\star = \rho\, \mathbb{E}_{\mu_{\lambda^\star+v^\star}}[g], \qquad \lambda_i^\star \, \mathbb{E}_{\mu_{\lambda^\star+v^\star}}[g_i] = 0, \;\; \mathbb{E}_{\mu_{\lambda^\star+v^\star}}[g_i] \leq 0 \;\; \forall i, \tag{14}$$

i.e., complementarity and feasibility for the moment constraints. Below we now state a practical algorithm that alternates (stochastic) dual ascent on $(\lambda, v)$ with LMC steps for the Gibbs tilt (10).

---

**Algorithm 1: AL-LMC**

**Data:** LMC step sizes $h_k > 0$; dual steps $\alpha_k > 0$ (for $\lambda$), $\eta_k^{(v)} > 0$ (for $v$); penalty $\rho > 0$.

    Initialize $\theta_0 \sim \mu_0$, $(\lambda_0, v_0) = (0, 0)$

**for** $k = 0, \dots, K-1$ **do**

    $\theta_{k+1} = \theta_k - h_k \, \nabla_\theta U(\theta_k; \lambda_k, v_k) + \sqrt{2h_k}\, \xi_k, \quad \xi_k \sim \mathcal{N}(0, I_d)$;

    $\widehat{g}_{k+1} \leftarrow$ estimate of $\mathbb{E}_{\mu_{\lambda_k+v_k}}[g(\theta)]$ (e.g., $g(\theta_{k+1})$ or empirical average);

    $v_{k+1} = v_k + \eta_k^{(v)} \big( \widehat{g}_{k+1} - \frac{1}{\rho} v_k \big)$;

    $\lambda_{k+1} = \max\big(0, \, \lambda_k + \alpha_k \widehat{g}_{k+1}\big)$;

**end**

---

## 4 Experiments

**Setup.** We use the *Adult* dataset Dua and Graff [2017], Chamon and Ribeiro [2020] to predict whether income exceeds $50k. The feature matrix has $N = 32{,}561$ examples and $d = 62$ features. We train a Bayesian logistic regression with independent zero-mean Gaussian priors of variance $\sigma^2 = 3$ on all coefficients. Let $\theta \in \mathbb{R}^d$ denote the parameters and $q(x; \theta) = \sigma(x^\top \theta)$ the positive-class probability. The posterior potential (binomial log-likelihood + Gaussian prior) is

$$f(\theta) = \sum_{n=1}^{N} \log\Big(1 + e^{-(2y_n - 1)\, x_n^\top \theta}\Big) \; + \; \sum_{i=0}^{d} \frac{\theta_i^2}{2\sigma^2}.$$

Gender is treated as the protected attribute. We enforce statistical parity across the two groups $\mathcal{G}_{\text{female}}$ and $\mathcal{G}_{\text{male}}$ with tolerance $\delta = 0.01$ by instantiating the empirical version of the constraint in (1). Equivalently, with $I = 2$ constraints on $\mathcal{G}_{\text{male}}$ and $\mathcal{G}_{\text{female}}$, we write the (empirical) constraint functions

$$\bar{g}_i(\theta) \; = \; \frac{1}{|\mathcal{G}_i|} \sum_{n \in \mathcal{G}_i} q(x_n; \theta) \; - \; \frac{1}{N} \sum_{n=1}^{N} q(x_n; \theta) \; - \; \delta_i$$

and require $\mathbb{E}_{\theta \sim \mu}[\bar{g}_i(\theta)] \geq 0$ for $i \in \{\text{female, male}\}$. We run the LMC kernel from with step size $h = \eta = 10^{-4}$ for $2 \times 10^4$ iterations and report statistics over the last $10^4$ samples; no constraints are imposed. We apply the fairness constraints using AL-LMC with step size $\eta_x = h = 10^{-4}$, with dual ascent step $\alpha = \eta_\lambda = 5 \times 10^{-3}$ and the $v$-update per Algorithm. 1.

**Results.** Under the unconstrained posterior $\pi$, the prevalence of positive predictions is $19.1\%$ overall, $26.2\%$ for males, and $0.05\%$ for females, amplifying the test-set disparity (male $30.6\%$ vs. female $10.9\%$). This behavior is visible across the distribution of prevalences under $\pi$ (Fig. 1). With the constrained sampler, prevalences become $17.1\%$ (overall), $18.1\%$ (male), and $15.1\%$ (female), substantially reducing the gap with an accuracy drop of only $\approx 2\%$. We observe substantial overlap between the male and female prevalence distributions under the constrained posterior $\mu^\star$. Dual trajectories indicate the constraint is primarily active on the female group: the male multiplier remains at 0 ($\lambda_{\text{male}} = 0$), consistent with the direction of correction.

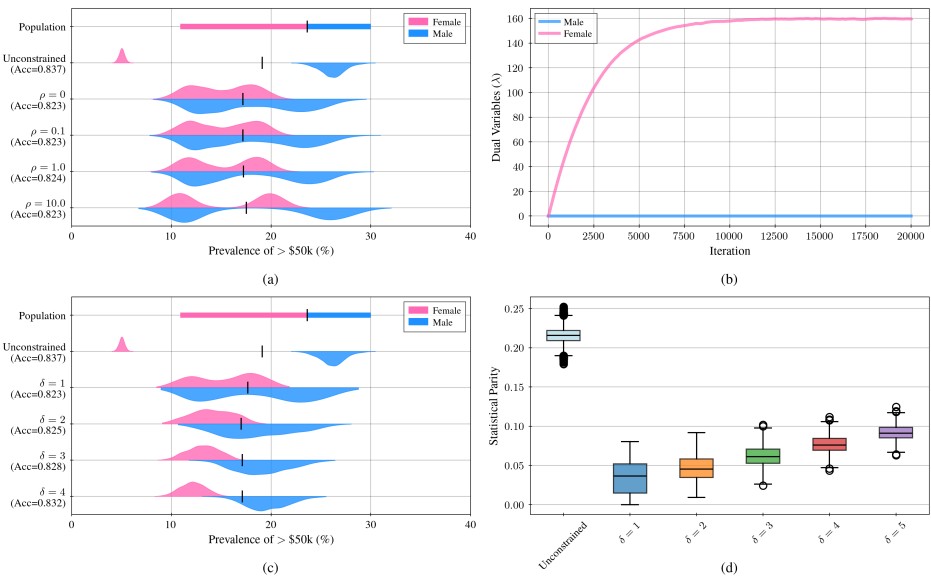

Figure 1: Comparison of fairness-constrained sampling methods on Adult ($N = 32{,}561$, $d = 62$) for Bayesian logistic regression predicting income $> \$50k$. **(a)** Prevalence comparison across penalty parameters $\rho$. We fix $\rho = 1.0$ for subsequent plots. **(b)** Dual variable trajectories of $\lambda_{\text{male}}$ and $\lambda_{\text{female}}$ over iterations; changes are guided entirely by $\lambda_{\text{female}}$. **(c)** Prevalence vs. $\delta$, larger $\delta$ relaxes the constraint. **(d)** Effect of $\delta$ on statistical parity, larger $\delta$ weakens the parity requirement.

## 5 Conclusion, Limitations and Future work

AL-LMC reliably enforces distributional rate constraints with small accuracy cost, but key challenges remain: principled, finite-sample tuning of the penalty $\rho$ and scaling to large/deep models via variance reduction and preconditioning. Future work should target adaptive $\rho$-schedules with provable guarantees and broader empirical/societal evaluation across fairness metrics. Extensions to other fairness notions (e.g., equalized odds), robustness under distribution shift, and $f$-divergence balls are important directions to pursue.

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

# A  Properties of the log partition function

$$F(\xi) := -\log Z(\xi), \qquad Z(\xi) := \mathbb{E}_\pi\big[e^{-\xi^\top g(\theta)}\big], \quad \xi \in \mathbb{R}^I. \tag{15}$$

**Assumptions**

A1. $g : \mathbb{R}^d \to \mathbb{R}^I$ is measurable and **bounded**: $|g(\theta)| \le G < \infty$ for all $\theta$.

These ensure $Z(\xi) \in (0, \infty)$ for all $\xi \in \mathbb{R}^I$, and justify differentiation under the integral via dominated convergence.

**Lemma 1.** *$F$ is $C^\infty$ and concave. Moreover,*

$$\nabla_\xi F(\xi) = \mathbb{E}_{\mu_\xi}[g(\theta)], \qquad \nabla_\xi^2 F(\xi) = -\operatorname{Cov}_{\mu_\xi}\big(g(\theta)\big) \preceq 0, \tag{16}$$

*where $\mu_\xi(\mathrm{d}\theta) = Z(\xi)^{-1}\pi(\theta)e^{-\xi^\top g(\theta)}\mathrm{d}\theta$.*

*Proof.* For each coordinate $j$,

$$\partial_{\xi_j} Z(\xi) = -\mathbb{E}_\pi\big[g_j(\theta)e^{-\xi^\top g(\theta)}\big], \qquad \partial_{\xi_i \xi_j}^2 Z(\xi) = \mathbb{E}_\pi\big[g_i(\theta)g_j(\theta)e^{-\xi^\top g(\theta)}\big]. \tag{17}$$

Dominated convergence applies because $|g_j(\theta)|e^{-\xi^\top g(\theta)} \le Ge^{|\xi||g(\theta)|} \le Ge^{|\xi|G}$, and similarly for the second derivative; both are integrable under $\pi$. By iterating this argument, all higher derivatives exist: $Z \in C^\infty$, and since the integrand is strictly positive, $Z(\xi) > 0$ for all $\xi$. Using $F = -\log Z$,

$$\nabla F(\xi) = -\frac{1}{Z(\xi)}\nabla Z(\xi) = -\frac{1}{Z(\xi)}\Big( -\mathbb{E}_\pi[g(\theta)e^{-\xi^\top g(\theta)}]\Big) = \mathbb{E}_{\mu_\xi}[g(\theta)]. \tag{18}$$

Differentiate again:

$$\nabla^2 F(\xi) = -\frac{1}{Z}\nabla^2 Z + \frac{1}{Z^2}(\nabla Z)(\nabla Z)^\top = -\Big(\mathbb{E}_{\mu_\xi}[gg^\top] - \mathbb{E}_{\mu_\xi}[g]\mathbb{E}_{\mu_\xi}[g]^\top\Big) = -\operatorname{Cov}_{\mu_\xi}(g). \tag{19}$$

The covariance matrix is positive semidefinite, hence $\nabla^2 F(\xi) \preceq 0$, so $F$ is concave.  □

**Lemma 2.** *Under $|g| \le G$, $\nabla F$ is globally Lipschitz with constant $L \le G^2$:*

$$|\nabla F(\xi) - \nabla F(\xi')| \le G^2|\xi - \xi'| \qquad \forall \xi, \xi'. \tag{20}$$

*Proof.* By the mean-value theorem in $\mathbb{R}^I$, for $\xi_t := \xi' + t(\xi - \xi')$,

$$\nabla F(\xi) - \nabla F(\xi') = \Big( \int_0^1 \nabla^2 F(\xi_t)dt\Big)(\xi - \xi'). \tag{21}$$

Thus

$$|\nabla F(\xi) - \nabla F(\xi')| \le \Big( \sup_{t \in [0,1]} |\nabla^2 F(\xi_t)| \Big) |\xi - \xi'| = \Big( \sup_t |\operatorname{Cov}_{\mu_{\xi_t}}(g)| \Big) |\xi - \xi'|. \tag{22}$$

For any probability $\nu$ and any unit vector $u$,

$$u^\top \operatorname{Cov}_\nu(g) u = \operatorname{Var}_\nu(u^\top g) \le \mathbb{E}_\nu[(u^\top g)^2] \le \mathbb{E}_\nu[|g|^2] \le G^2. \tag{23}$$

Taking the supremum over $u$ yields $|\operatorname{Cov}_\nu(g)| \le G^2$. Hence the Lipschitz constant is at most $G^2$. $\qquad\square$

**Lemma 3.** *Under $|g| \le G$, $Z(\xi) \in (0, \infty)$ for all $\xi \in \mathbb{R}^I$; in particular $F(\xi)$ is finite everywhere.*

*Proof.* For any $\theta$, $|\xi^\top g(\theta)| \le |\xi||g(\theta)| \le |\xi|G$. Hence

$$e^{-|\xi|G} \le e^{-\xi^\top g(\theta)} \le e^{|\xi|G}. \tag{24}$$

Integrating under $\pi$ (a probability measure) gives

$$e^{-|\xi|G} \le Z(\xi) \le e^{|\xi|G}, \tag{25}$$

so $Z(\xi)$ is strictly positive and finite for all $\xi$. $\qquad\square$

# B  Properties of the dual in $v$

We fix $\lambda \in \mathbb{R}_+^I$ throughout and study

$$\phi_\lambda(v) := -\tfrac{1}{2\rho}|v|^2 + F(\lambda + v), \qquad \rho > 0, \tag{26}$$

where (from part A) $F(\xi) = -\log \mathbb{E}_\pi[e^{-\xi^\top g(\theta)}]$.

By the previous section, $F \in C^\infty$, $\nabla F(\xi) = \mathbb{E}_{\mu_\xi}[g]$, and $\nabla^2 F(\xi) = -\operatorname{Cov}_{\mu_\xi}(g) \preceq 0$ for all $\xi$.

**Lemma 4.** *$\phi_\lambda$ is $(1/\rho)$-strongly concave in $v$ and has $L_v$-Lipschitz gradient with $L_v \le 1/\rho + G^2$. Hence the maximizer $v^\star(\lambda)$ is unique.*

*Proof.* Differentiate:

$$\nabla_v \phi_\lambda(v) = -\tfrac{1}{\rho}v + \nabla F(\lambda + v), \qquad \nabla_v^2 \phi_\lambda(v) = -\tfrac{1}{\rho}I + \nabla^2 F(\lambda + v). \tag{27}$$

By B2, $\nabla^2 F(\cdot) \preceq 0$, so $\nabla_v^2 \phi_\lambda(v) \preceq -\tfrac{1}{\rho}I$ for all $v$. This is exactly $(1/\rho)$-strong concavity. Strong concavity implies a unique maximizer.

For the gradient Lipschitz constant,

$$|\nabla_v^2 \phi_\lambda(v)| \le \tfrac{1}{\rho} + |\nabla^2 F(\lambda + v)| \le \tfrac{1}{\rho} + |\operatorname{Cov}_{\mu_{\lambda+v}}(g)| \le \tfrac{1}{\rho} + G^2, \tag{28}$$

since for any covariance matrix $|\operatorname{Cov}(g)| \le \mathbb{E}|g|^2 \le G^2$. Thus $\nabla \phi_\lambda$ is $L_v$-Lipschitz with $L_v \le 1/\rho + G^2$. $\qquad\square$

**Lemma 5.** *Stationarity of $\phi_\lambda$ is equivalent to*

$$v = \rho m(\lambda + v), \qquad m(\xi) := \nabla F(\xi) = \mathbb{E}_{\mu_\xi}[g], \tag{29}$$

*so $v^\star(\lambda)$ is a fixed point of $T(v) := \rho m(\lambda + v)$. Moreover,*

$$|T(v) - T(v')| \le \rho|m(\lambda + v) - m(\lambda + v')| \le \rho G^2 |v - v'|, \tag{30}$$

*hence if $\rho G^2 < 1$ the map $T$ is a contraction and the fixed point is unique with linear convergence ($|v_{t+1} - v^\star| \le (\rho G^2)|v_t - v^\star|$ for the iteration $v_{t+1} = T(v_t)$).*

*Proof.* First-order optimality gives $\nabla_v \phi_\lambda(v) = 0 \iff -v/\rho + \nabla F(\lambda + v) = 0 \iff v = \rho m(\lambda + v)$. Thus $v^\star$ is a fixed point of $T$.

By the global Lipschitzness of $\nabla F$, $m(\cdot) = \nabla F(\cdot)$ is Lipschitz with constant $\le G^2$. Therefore

$$|T(v) - T(v')| = \rho|m(\lambda + v) - m(\lambda + v')| \le \rho G^2 |v - v'|. \tag{31}$$

If $\rho G^2 < 1$, $T$ is a contraction on $\mathbb{R}^I$, so Banach's fixed-point theorem yields uniqueness and linear convergence of the fixed-point iteration. $\qquad\square$

**Lemma 6.** *Any maximizer $v^\star(\lambda)$ satisfies $|v^\star(\lambda)| \leq \rho G$.*

*Proof.* From stationarity: $v^\star = \rho \mathbb{E}_{\mu_{\lambda+v^\star}}[g]$. Then

$$|v^\star| = \rho |\mathbb{E}_{\mu_{\lambda+v^\star}}[g]| \leq \rho \mathbb{E}_{\mu_{\lambda+v^\star}}[|g|] \leq \rho G. \qquad (32)$$

$\square$

**Lemma 7.** *If you run exact gradient ascent*

$$v_{t+1} = v_t + \eta_v \nabla_v \phi_\lambda(v_t), \qquad (33)$$

*with any step size $\eta_v \in (0, 1/L_v]$ (where $L_v \leq 1/\rho + G^2$ from the strong concavity lemma), then*

$$\phi_\lambda(v_{t+1}) \geq \phi_\lambda(v_t) + \tfrac{\eta_v}{2}|\nabla_v \phi_\lambda(v_t)|^2 \geq \phi_\lambda(v_t), \qquad (34)$$

*so the dual objective increases monotonically.*

*Proof.* Since $\phi_\lambda$ has $L_v$-Lipschitz gradient, the standard smoothness inequality for concave functions gives (apply the convex version to $-\phi_\lambda$)

$$\phi_\lambda(v_{t+1}) \geq \phi_\lambda(v_t) + \nabla \phi_\lambda(v_t)^\top (v_{t+1} - v_t) - \tfrac{L_v}{2}|v_{t+1} - v_t|^2. \qquad (35)$$

With $v_{t+1} - v_t = \eta_v \nabla \phi_\lambda(v_t)$ and $\eta_v \leq 1/L_v$,

$$\phi_\lambda(v_{t+1}) \geq \phi_\lambda(v_t) + \eta_v |\nabla \phi_\lambda(v_t)|^2 - \tfrac{L_v}{2}\eta_v^2 |\nabla \phi_\lambda(v_t)|^2 \geq \phi_\lambda(v_t) + \tfrac{\eta_v}{2}|\nabla \phi_\lambda(v_t)|^2. \qquad (36)$$

$\square$

**Lemma 8.** *If, in addition, $\phi_\lambda$ is $\mu_v$-strongly concave with $\mu_v = 1/\rho$, then taking $\eta_v = 1/L_v$ yields*

$$|v_{t+1} - v^\star(\lambda)| \leq \left(1 - \frac{\mu_v}{L_v}\right)|v_t - v^\star(\lambda)| \leq \left(1 - \frac{1/\rho}{1/\rho + G^2}\right)|v_t - v^\star(\lambda)| = \frac{\rho G^2}{1 + \rho G^2}|v_t - v^\star(\lambda)|. \qquad (37)$$

*Proof.* For $\mu_v$-strongly concave and $L_v$-smooth $\phi_\lambda$, the gradient mapping for ascent with step $1/L_v$ is a contraction with factor $1 - \mu_v/L_v$ in the Euclidean norm (the standard result dual to strong convexity + smoothness for gradient descent). Concretely, letting $q(v) := -\phi_\lambda(v)$, $q$ is $L_v$-smooth and $\mu_v$-strongly convex, so gradient descent on $q$ with step $1/L_v$ satisfies

$$|v_{t+1} - v^\star| \leq \left(1 - \tfrac{\mu_v}{L_v}\right)|v_t - v^\star|. \qquad (38)$$

Switching back to ascent on $\phi_\lambda$ gives the stated bound, and substituting $\mu_v = 1/\rho$ plus $L_v \leq 1/\rho + G^2$ yields the explicit factor $\rho G^2/(1 + \rho G^2) < 1$. $\square$

