# OpenReview forum: "Augmented Lagrangian Langevin Monte Carlo for Fair Inference"
_EurIPS.cc/2025/Workshop/UPLB — UPLB2025_

### Official Review · Reviewer_EpcS · 2025-10-27
**Useful sampling algorithm**

**Rating:** 7
**Confidence:** 3

**Review:**

This paper suggests a primal-dual sampler enforcing expectation-type constraints on the posterior.
The mathematical derivations appear to be correct and the suggested algorithm viable.
A major limitation is the lack of principle for choosing the penalty parameter $\rho$. But the authors are aware of it.
Apart form this, I only have a few minor comments:
1) In the regression, I suppse "gender" has been excluded from the features. Please be explicit about it.
2) $\tilde\pi$, in eqs. (5) and (6) should be $\pi$, correct?
3) In line 98, I suppose the constraint should read $\leq 0$, in line with Sect. 2?

---

### Decision · Program_Chairs · 2025-11-03

Accept (Poster)